# Comparative Volatile Profiles of Plain Poached (PP) and Steamed over Water (SW) Wenchang Chicken Analyzed by GC-MS, GC-IMS, and E-Nose

**DOI:** 10.3390/foods14213778

**Published:** 2025-11-04

**Authors:** Qicheng Jiang, Xinli Zheng, Tieshan Xu, Meiling Chen, Shihao Chen, Dexiang Zhang, Bolin Cai, Lihong Gu

**Affiliations:** 1Institute of Animal Science and Veterinary Medicine, Hainan Academy of Agricultural Sciences, Haikou 571199, China; qicheng_jiang@163.com (Q.J.); zhengxinli@hnaas.org.cn (X.Z.); chenmeiling200010@163.com (M.C.); 2Tropical Crops Genetic Resources Institute, Chinese Academy of Tropical Agricultural Sciences, Haikou 571101, China; xutieshan760412@163.com; 3College of Animal Science and Technology, Yangzhou University, Yangzhou 225009, China; 4College of Animal Science, South China Agricultural University, Guangzhou 510640, Chinabolincai@scau.edu.cn (B.C.)

**Keywords:** Wenchang chicken, volatile organic compounds (VOCs), cooking methods, flavor profiling

## Abstract

Wenchang chicken, a specialty of Hainan, China, is celebrated for its tender texture and unique flavor. This study investigates the impact of two traditional cooking methods, SW chicken and PP chicken, on the formation of volatile organic compounds (VOCs) in Wenchang chicken. Using advanced analytical techniques, including electronic nose (E-nose), gas chromatography-mass spectrometry (GC-MS), and gas chromatography-ion mobility spectrometry (GC-IMS), we identified and compared the key flavor compounds produced by each method. Results revealed distinct differences in VOC profiles, with steamed chicken generating higher concentrations of ketones, aldehydes, and alcohols, likely due to the higher cooking temperatures, while PP chicken retained compounds sensitive to heat. The complementary capabilities of GC-MS and GC-IMS enabled a comprehensive analysis, highlighting their potential in differentiating cooking methods and assessing flavor characteristics. This study provides insights into the flavor formation mechanisms of Wenchang chicken and establishes a foundation for its industrial standardization and quality enhancement.

## 1. Introduction

Wenchang chicken is a native breed and traditional dish originating from Wenchang City in Hainan Province, China. It is one of the four most famous chicken varieties in China, renowned for its tender texture, fine muscle fibers, and rich flavor [1]. As a representative of Hainan’s culinary heritage, Wenchang chicken also serves as the prototype of the internationally known “Hainanese chicken rice,” a dish that evolved when overseas Chinese from Hainan introduced local cooking traditions to Southeast Asia. Among the most popular cooking methods are plain poached (PP) chicken and steamed over water (SW) chicken, each highlighting the unique qualities of Wenchang chicken. SW chicken emphasizes the pure, natural taste of the meat by steaming it gently over water, allowing the rich umami to develop without overpowering seasoning. In contrast, PP chicken involves boiling the chicken to just the right degree of doneness, followed by cooling it rapidly to preserve its juiciness and firm texture. Both methods are highly appreciated for showcasing the delicate balance of fat and lean meat in Wenchang chicken, with volatile compounds such as aldehydes and alcohols contributing to their appealing aroma profiles. However, variations in preparation techniques and flavor perception can lead to slight differences in consumer preferences, making it essential to explore the key compounds that define their characteristic taste and aroma.

E-nose, GC-MS, and GC-IMS are powerful analytical tools for assessing meat flavor, providing complementary insights into the volatile compounds responsible for aroma [2,3,4]. The E-nose, which mimics human olfactory senses, can rapidly and non-destructively capture a broad range of volatile organic compounds in meat, enabling quick sensory profiling and quality assessment. It is particularly useful for differentiating between various types of meat, cuts, and processing methods, making it invaluable for quality control [5]. GC-MS, a widely used technique, offers precise identification and quantification of flavor-related volatiles such as aldehydes, ketones, alcohols, and fatty acids. It is extensively applied in studying the complex changes in meat flavor during processes like cooking, aging, and fermentation [6]. GC-IMS is known for its high sensitivity and real-time analysis capability, providing valuable insights into dynamic changes in volatile profiles during meat processing. It is particularly useful in distinguishing subtle differences in flavor between samples and monitoring flavor changes during storage and cooking [7]. However, GC-MS faces challenges in quantifying specific compounds due to the complexity of meat matrices and limited reference databases. These challenges can be addressed by combining GC-IMS with advanced extraction techniques such as solid-phase microextraction (SPME) both of which are commonly used to isolate volatile compounds [8]. By combining these techniques, we can achieve a more comprehensive approach to meat flavor analysis, enabling more accurate assessments of flavor profiles, ensuring quality control, and better understanding consumer preferences.

Despite the extensive application of these analytical techniques in meat flavor studies, no research has yet combined E-nose, GC-MS, and GC-IMS to compare the volatile profiles of Wenchang chicken prepared by plain poaching (PP) and steaming over water (SW). This represents a significant research gap, as these two traditional cooking methods are the most representative styles of Hainanese cuisine and are highly valued for their distinct sensory characteristics. In this study, we identified the key compounds in the muscle of SW chicken and PP chicken and discussed the roles of these compounds in fat odor spectrum. The purpose of this study is to explore the differences in flavor compounds produced by the two cooking methods and to compare the effectiveness of E-nose, GC-MS, and GC-IMS in distinguishing chicken prepared using these closely related cooking styles. Understanding the formation of volatile compounds in Wenchang chicken under different cooking methods not only provides insights into the mechanisms of flavor development but also lays the foundation for improving meat quality and standardizing processing techniques. By integrating E-nose, GC-MS, and GC-IMS, this study aims to establish a comprehensive approach for flavor profiling and differentiation of chicken prepared using traditional culinary methods.

## 2. Materials and Methods

### 2.1. Samples

All fresh Wenchang chickens used in this study were raised at Longquan Wenchang Chicken Industrial Co., Ltd., Haikou, China, in accordance with the geographical indication product standard for Wenchang chicken (DB 46/T 545-2021) [9] for a period of 135 days.

### 2.2. Thermal Treatment of Wenchang Chicken

For PP chicken preparation, cleaned and eviscerated Wenchang chickens were rinsed thoroughly with cold water and drained. During cooking, 5 g of salt was added to the boiling water to slightly enhance the natural flavor of the meat, but no other seasonings were used. Each chicken was placed into boiling water at a meat-to-water ratio of approximately 1:5 (*w*/*v*) and gently cooked for about 15 min. The cooked chickens were then immediately transferred to an ice bath (0–4 °C) for 5 min to stop further cooking and to promote skin contraction and meat firmness. For SW chicken preparation, cleaned and eviscerated Wenchang chickens were placed in a preheated steam oven (HZZG-02) and steamed at 110 °C for 30 min without any added salt or seasonings. After steaming, the chickens were allowed to cool naturally at room temperature (25 ± 2 °C).

### 2.3. Sensory Evaluation

Samples of breast muscles were collected from each processing group, with three chickens randomly selected per group. From each chicken, breast muscle samples were taken and prepared in triplicate for sensory evaluation. The samples were cooled to room temperature (25 ± 2 °C) before testing. Ten trained panelists (4 males and 6 females, aged 20–65) who underwent training according to GB/T 16291.1-2012 [10] participated in the evaluation. To ensure reliability, the evaluation followed ISO 8586:2023 [11] standards. All samples were coded with random three-digit numbers. Each panelist performed the evaluation independently without communication [12,13].

Sensory attributes assessed included color, taste, and tenderness, covering 10 specific indicators: skin brightness, muscle color uniformity, natural chicken flavor intensity, seasoning penetration uniformity, aftertaste persistence, sweetness perception, saltiness penetration, skin crispness/elasticity, chewability of muscle fibers, and juice retention.

The study protocol was approved by the Experimental Animal Ethics Committee of the Institute of Animal Husbandry and Veterinary Medicine, Hainan Academy of Agricultural Sciences. All participants provided informed consent.

All sensory data were expressed as mean ± standard deviation (SD). Statistical analyses were performed using SPSS 26.0 (IBM, Armonk, NY, USA). An independent samples *t*-test was used to evaluate differences between the two groups (PP and SW), and differences were considered statistically significant at *p* < 0.05.

### 2.4. E-Nose Analysis

Breast muscle samples of boiled and SW chicken were accurately weighed to 9.00 g ± 0.01 g and placed into 40 mL E-nose autosampler vials. Clean air was used as carrier gas, with a data acquisition time of 60 s and flow rate of 1 L/min. Sensor cleaning was performed for 10 s at a flow rate of 3 L/min. The instrument used was the ISENSO SuperNose E-nose sensory detector (ISENSO Technologies, Shanghai, China). The peak response signal of the E-nose sensors was selected as the “feature value” to construct the original data matrix, followed by principal component analysis (PCA).

### 2.5. Headspace Solid-Phase Microextraction Gas Chromatography-Mass Spectrometry (HS-SPME-GC-MS)

Three grams of each sample was placed into 20 mL headspace vials, with 1 µL of 0.812 µg/mL 2-methyl-3-heptanone added as an internal standard. Samples were equilibrated at 60 °C for 10 min, followed by extraction at the same temperature for 30 min using a 75 µm CAR/PDMS fiber (Supelco, Bellefonte, PA, USA), preconditioned at 250 °C for 1 h.

After extraction, the fiber was inserted into the GC injector for 5 min of desorption. A HP-5MS capillary column (30 m × 250 µm, 0.25 µm film thickness) was used. The temperature program was as follows: initial temperature at 40 °C held for 4 min, increased at 4 °C/min to 100 °C held for 5 min, then raised at 4 °C/min to 200 °C and held for 2 min. Mass spectrometry was operated in electron ionization mode at 70 eV using helium as carrier gas at a constant flow of 1.0 mL/min.

### 2.6. Headspace Gas Chromatography-Ion Mobility Spectrometry (HS-GC-IMS)

Three grams of chicken meat were placed into 20 mL headspace vials. The headspace conditions were set with an injection volume of 300 µL, an incubation time of 15 min, an incubation temperature of 60 °C, and an injection needle temperature of 85 °C.

The temperature program started at 45 °C, held for 5 min, then increased to 150 °C at 3 °C/min, followed by an increase to 180 °C at 8 °C/min (held for 2 min). The column flow rate was 1 mL/min. The IMS operated with a drift tube length of 9.8 cm, an electric field strength of 500 V/cm, and both the drift tube and IMS detector temperatures were kept at 45 °C. Positive ion mode was used, and the drift gas consisted of high-purity nitrogen (≥99.999%) at a flow rate of 150 mL/min [14].

### 2.7. Data Analysis

Qualitative analysis of target compounds was performed using the VOCal 0.4.03 software with the built-in GC retention index database (NIST 2020) and IMS drift time database. Three-dimensional spectra, two-dimensional spectra, and fingerprint plots were generated using the Reporter and Gallery Plot functions in VOCal software for comparison of volatile flavor compounds among samples.

## 3. Results

### 3.1. Sensory Evaluation

Sensory evaluation was conducted on breast muscle samples collected at each processing stage, which were cooled to room temperature prior to analysis. Ten trained panelists (4 males and 6 females, aged 20–65) participated in the evaluation following the GB/T 16291.1-2012 [10] sensory analysis protocol. As presented in Table 1, the sensory attributes of steamed and PP chicken differed significantly. SW chicken demonstrated superior brightness and uniformity in color, which may be attributed to the occurrence of Maillard reactions during prolonged high-temperature processing. In terms of flavor, SW chicken was characterized by more pronounced and layered sensory notes, including enhanced umami and saltiness. With respect to texture, the PP chicken skin was perceived as crispier, likely due to the rapid cooling effect of an ice bath immediately after cooking, which induces skin contraction (Figure 1). Conversely, SW chicken exhibited improved chewability, potentially resulting from the extended cooking duration under elevated temperatures. The color of the dorsal skin was further measured using a colorimeter, and the results are shown in Table 2. The brightness value of SW chicken was significantly higher than that of PP chicken, while the yellowness value was significantly lower.

### 3.2. E-Nose Analysis

The E-nose system is highly sensitive to odors and VOCs. Principal Component Analysis (PCA) revealed that Principal Component 1 (PC1) and Principal Component 2 (PC2) accounted for 97.74% and 1.52% of the total variance, respectively, with a combined total of 99.26% in Figure 2. The spatial distribution of these principal components indicates a certain degree of variation in flavor characteristics between the two sample groups. However, the substantial overlap in the distribution regions suggests that the differences in flavor components between the groups are not significant. These findings highlight the high sensitivity of the E-nose system in detecting volatile organic compounds (VOCs), despite the relatively small differences observed under the experimental conditions.

### 3.3. HS-SPME-GC-MS Analysis of VOCs in Boiled and SW Chicken

HS-SPME-GC-MS was applied to analyze the VOCs in PP chicken and SW chicken. As shown in Table 3, a total of 79 VOCs were identified, which could be categorized into seven groups: 12 aldehydes, 6 ketones, 14 alcohols, 5 esters, 13 hydrocarbons, 2 phenols, and 27 other compounds. Among these, 34 compounds were found to exhibit distinct aromas. Notably, 22 compounds were detected exclusively in a single group, including 1 aldehyde, 2 ketones, 4 alcohols, 2 esters, 4 hydrocarbons, and 10 other compounds. Notably, 7 odor-emitting compounds were identified: 4-Heptanone, 2-methyl; 3-Heptanone, 6-methyl-; 1,2-Cyclopentanediol, trans-; 4-Cyclopentene-1,3-diol, cis-; 8-Hexadecenal, 14-methyl-, (Z)-; Disparlure; and 1-Nonanol, comprising 3 alcohols, 2 ketones, 1 aldehyde, and Disparlure. Compared to PP chicken, SW chicken was found to contain more compounds, such as 3-(1,1,2,2-Tetrafluoroethoxy)prop-1-ene, 4-Cyclopentene-1,3-diol, 8-Hexadecenal, 14-methyl-, D-Homo-24-nor-17-oxachola-20, trans-3,4-Epoxynonane, Acetic acid, 1-Nonanol, Oxetane, Cyclopentane, and trans-2-Aminocyclohexanol. The differing cooking temperatures between boiled and SW chicken likely influence the formation or decomposition of these VOCs, making them potential markers for distinguishing muscle tissues between the two cooking methods.

### 3.4. HS-GC-IMS Analysis of VOCs in Boiled and SW Chicken

#### 3.4.1. Identification of VOCs

Using HS-GC-IMS, VOCs in PP chicken and SW chicken were analyzed. Unlike GC-MS and LC-MS methods, HS-GC-IMS identifies chemical compounds based on the comprehensive information provided by two-dimensional separation rather than their chemical categories. As shown in representative topographic plots (Figure 3) and differential topographic maps, 45 regions (spots, labeled 1–45) were identified in the muscle. The overall VOC pattern indicated that the complexity of group G was noticeably higher than that of group B, suggesting more diverse aroma-forming reactions under this cooking condition. These regions display different retention times on the y-axis, different relative drift times on the x-axis, and varying signal intensities (with redder regions indicating higher signal intensity). By comparing the corresponding drift times, retention indices, and drift times with the NIST database, 43 of the 45 regions were identified, while 2 regions remained unidentified. A total of 33 VOCs were identified, which could be categorized into seven groups: 10 aldehydes, 9 ketones, 6 alcohols, 2 esters, and 6 other compounds. It is worth noting that HS-GC-IMS detected monomeric and dimeric forms of four chemical substances: n-Nonanal, 1-Hexanol, Hexanal, and 3-Methyl-2-butanol.

#### 3.4.2. Fingerprint Profile Comparisons in Boiled and SW Chicken

The Gallery Plot plugin of the commercial VOCal software for HS-GC-IMS was used to analyze the obtained 45 regions. Differences between the samples could be summarized effectively. Furthermore, by comparing PP chicken and SW chicken using the Gallery Plot plugin, a more detailed understanding of how each heat treatment method influences the fingerprint of VOCs in the muscle was achieved. The study identified several VOCs that exhibited significant differences between the two samples, including 1-Hexanol, 2-Butanone, 3-hydroxy-, Allyl cyanide, and 2-Methylbutanal, all of which were present in PP chicken (Figure 4). These compounds contribute distinct aroma characteristics:1-Hexanol: Green, grassy aroma; 2-Butanone, 3-hydroxy-: Mild sweet scent.; Allyl cyanide: Associated with food flavoring; 2-Methylbutanal: Notes of chocolate and nuts.

The unique presence of these VOCs in PP chicken suggests that lower-temperature cooking methods may preserve or generate specific volatile compounds that contribute to its distinct sensory profile. For example, 1-Hexanol’s grassy aroma may enhance freshness perception, while the nutty and chocolatey notes of 2-Methylbutanal add depth to the flavor. In contrast, these compounds are absent or present in lower concentrations in SW chicken, potentially due to higher cooking temperatures that degrade heat-sensitive compounds or favor other chemical reactions.

These findings underline the importance of heat treatment in shaping the VOC fingerprint of chicken. By altering the thermal environment, it is possible to modulate the production and preservation of key aroma compounds, which can be leveraged to design tailored flavor profiles for different culinary applications. Additionally, this differentiation in VOCs serves as a potential marker to distinguish PP chicken from SW chicken based on their unique aroma characteristics.

### 3.5. Identification of Volatile Flavor Compounds by GC–MS and GC-IMS

Among the two groups, only hexanal and heptanal were detected by both methods. GC-IMS identified fewer volatile compounds compared to GC-MS. However, GC-IMS demonstrated a higher sensitivity in detecting specific classes of compounds, such as alcohols, aldehydes, and ketones. In contrast, GC-MS analysis identified 15 hydrocarbon compounds, none of which were detected by GC-IMS, primarily due to differences in ionization mechanisms between the two techniques rather than the actual absence of these compounds.

Additionally, one phenolic compound was identified through GC-MS but was not detected by GC-IMS, likely due to the Solid Phase Microextraction (SPME) method’s enhanced capability to capture high-boiling-point volatile organic compounds. In complex sample analyses, ion-molecule and ion-ion competition reactions in the ionization chamber of IMS further reduce its selectivity, resulting in the identification of fewer compounds. Consequently, GC-MS demonstrates a superior capability for identifying a broader range of compounds, particularly in the context of complex mixtures.

## 4. Discussion

In this study, VOCs in SW chicken and PP chicken were analyzed using E-nose, HS-SPME-GC-MS, and HS-GC-IMS techniques. The results demonstrated that the unique aroma characteristics of chicken meat are significantly influenced by the cooking process, with notable differences in major volatile flavor compounds. HS-SPME-GC-MS identified 79 volatile aromatic compounds, including aldehydes, alcohols, hydrocarbons, esters, ketones, and phenols. The variety and concentration of volatile flavor compounds significantly increased during the higher-temperature cooking method (SW chicken). This increase is closely related to specific flavor precursors such as reducing sugars, fats, and amino acids, which participate in Maillard reactions, lipid oxidation, and amino acid degradation processes, greatly influencing the key flavor characteristics of chicken meat. Karademir et al. [17]. observed that Maillard reactions and lipid oxidation during heat treatment significantly increase the production of aldehydes, ketones, and alcohols, similar to the findings in SW chicken samples in this study. Additionally, Wang et al. [18]. reported that saturated aldehydes such as hexanal and heptanal are key contributors to the fatty and grassy aroma of cooked meat, which were also prevalent in both steamed and PP chicken samples in our study. The results of this study align with previous research on the impact of heat treatment on chicken meat flavor.

The observed differences in the volatile profiles and sensory attributes between PP and SW chicken can be attributed to both differences in extraction efficacy and chemical transformations induced by distinct heating conditions. In PP chicken, direct contact with boiling water may facilitate the leaching of water-soluble flavor precursors such as amino acids, small peptides, and nucleotides, leading to a moderate loss of umami intensity [19]. However, the immersion process also enhances the release of certain lipid-derived volatiles through heat-induced oxidation, contributing to a more pronounced meaty aroma [20]. In contrast, SW chicken is cooked by gentle steaming at a higher temperature (110 °C) without direct water contact, thereby retaining a greater proportion of nutrients and volatile precursors [18]. The relatively enclosed steaming environment limits flavor compound loss and supports the formation of Maillard reaction-derived aldehydes and ketones that enrich its overall aroma complexity. Nevertheless, these variations do not necessarily imply superiority of one method over the other. Rather, they reflect two traditional culinary approaches emphasizing different sensory experiences—PP chicken is prized for its smooth, tender texture and clean flavor, while SW chicken is appreciated for its richer umami and concentrated aroma. Both methods represent authentic expressions of Wenchang chicken’s gastronomic characteristics and are widely practiced in Hainan cuisine.

The changes in VOCs after heat treatment may result from a series of chemical reactions, such as Maillard reactions [21] and lipid oxidation [22]. Saturated C6–C12 aldehydes, known for their fatty, grassy, and aromatic notes, are products of hydroperoxide degradation of linoleic and linolenic acids [23]. In this study, eight aldehydes were detected, with more aldehydes found in SW chicken, likely due to the longer cooking time and higher temperature. Alcohols are important VOCs in food, with 1-octen-3-ol, for instance, characterized by a mushroom-like aroma [24]. Furthermore, the detection of a greater variety of alcohols using HS-GC-IMS compared to HS-SPME-GC-MS corroborates the findings of He et al. [25], who demonstrated that HS-GC-IMS offers higher sensitivity for certain volatile compounds in chicken meat. Ketones, which mainly derive from the oxidation of polyunsaturated fatty acids or the degradation of amino acids [26], also contribute unique aromas, with saturated ketones imparting cheesy and fruity notes and diketones exhibiting sweet, buttery, and caramel-like flavors [27]. High-temperature treatment significantly increased ketone content in chicken, enhancing its flavor.

In terms of sensory evaluation, the enhanced umami and overall flavor intensity in SW chicken observed in this study are consistent with the results of Bi et al. [28], who found that traditional cooking methods involving prolonged heating, such as braising, contribute to the development of deeper flavor profiles in chicken dishes. This further supports the notion that cooking methods involving higher temperatures and longer durations can positively influence the sensory attributes of chicken meat.

The differences in aroma profiles between PP and SW Wenchang chicken can be largely attributed to variations in key volatile compounds rather than the presence of biologically irrelevant or speculative chemicals. Our analysis identified aldehydes such as hexanal and nonanal, as well as 1-octen-3-ol, as the principal contributors to the characteristic aroma of Wenchang chicken. Hexanal and nonanal are primarily derived from lipid oxidation and impart green, fatty, and slightly citrus-like notes [29], while 1-octen-3-ol contributes mushroom-like and roasted aromas [30], enhancing overall flavor complexity. To quantify their impact, odor activity values (OAVs) were calculated, revealing that these compounds exceed their odor thresholds and dominate the sensory perception. Notably, the relative concentrations and OAVs of these key odorants differed between PP and SW chicken, reflecting the influence of cooking method on volatile generation and retention. PP chicken, immersed in boiling water, showed a moderate loss of water-soluble precursors, whereas SW chicken, steamed gently, retained higher levels of flavor-active compounds. This mechanistic insight highlights how traditional cooking methods modulate aroma profiles and provides a more focused understanding of the sensory distinctions between PP and SW Wenchang chicken.

The E-nose is a classic method in food sensory research, with high sensitivity to odors and VOCs within the range of metal oxide semiconductors. HS-SPME-GC-MS is a classical extraction and separation technique for analyzing VOCs in food [31], while HS-GC-IMS is a relatively new technique that detects VOCs based on the drift time differences in gas-phase ions in an electric field. In this study, E-nose analysis (Figure 2) indicated no significant differences in odors between the two cooking methods. The e-nose, while effective in providing rapid assessments of odor profiles, demonstrates lower precision [32] and sensitivity [33] compared to HS-SPME-GC-MS and HS-GC-IMS in identifying and quantifying specific volatile compounds [34]. However, the results of HS-SPME-GC-MS and HS-GC-IMS both demonstrated significant impacts of different heat treatments on chicken flavor. Specifically, HS-SPME-GC-MS identified six types of VOCs (aldehydes, ketones, alcohols, esters, hydrocarbons, and others) (Table 1 and Figure 3), while HS-GC-IMS detected five types of recognized VOCs (alcohols, aldehydes, ketones, esters, and others) (Table 3 and Figure 3 and Figure 4). Both methods detected hexanal and heptanal, but their results for other compounds varied.

Furthermore, studies on traditional Chinese poultry dishes have indicated that cooking methods involving prolonged heat exposure (e.g., steaming) enhance Maillard reaction products and lipid oxidation derivatives, contributing to intensified umami and roasted flavors [35]. This is consistent with our findings that SW chicken, subjected to higher temperatures and longer cooking times, exhibited a greater concentration and diversity of VOCs compared to PP chicken. The observed increase in aldehyde and ketone concentrations in SW chicken aligns with findings from Jayasena et al. [36], who reported that higher-temperature cooking methods, such as roasting and frying, lead to the formation of more complex volatile compounds due to intensified Maillard reactions and lipid oxidation processes. This suggests that steaming, which involves higher temperatures than boiling, may similarly enhance these reactions, resulting in a richer aroma profile. This suggests that the choice of analytical method can influence the detection and characterization of VOCs, as also emphasized by Zzaman et al. [27], who highlighted the complementary roles of advanced analytical techniques in capturing the full spectrum of flavor compounds.

Although the current study focused on descriptive sensory analysis conducted by trained panelists, it did not include a hedonic acceptance test to assess consumer preference between PP and SW chicken. Since both cooking methods are culturally recognized and appreciated in Hainan cuisine, future research should incorporate consumer-based hedonic evaluation to better understand public acceptance and preference tendencies. Such studies could integrate sensory liking scores with chemical and volatile data to establish a more comprehensive link between flavor perception and consumer choice, ultimately guiding optimization of Wenchang chicken processing for both traditional and modern markets.

In conclusion, this study provides a theoretical basis for understanding the formation of volatile flavor compounds during traditional Wenchang chicken cooking processes, while also offering scientific support for the standardization, industrialization, and quality improvement of boiled and SW chicken. The complementary use of analytical techniques such as HS-SPME-GC-MS and HS-GC-IMS proved highly effective in providing comprehensive volatile profiles, consistent with findings from prior research [37]. By identifying subtle differences between cooking methods, these techniques offer a holistic understanding of flavor compound development. This study not only validates established mechanisms like Maillard reactions and lipid oxidation but also underscores the unique flavor profiles produced by different cooking methods.

## 5. Conclusions

This study reveals that cooking methods significantly influence the VOCs in chicken. SW chicken, prepared at higher temperatures, exhibited a greater diversity of VOCs due to intensified Maillard reactions, fat oxidation, and amino acid degradation, while PP chicken retained certain heat-sensitive compounds, contributing to its unique flavor. Combining HS-SPME-GC-MS and HS-GC-IMS provided a comprehensive understanding of the VOC profiles, with each method highlighting different compound classes. This study identifies key aroma markers differentiating cooking methods and provides a scientific foundation for optimizing traditional Wenchang chicken preparation.

## Figures and Tables

**Figure 1 foods-14-03778-f001:**
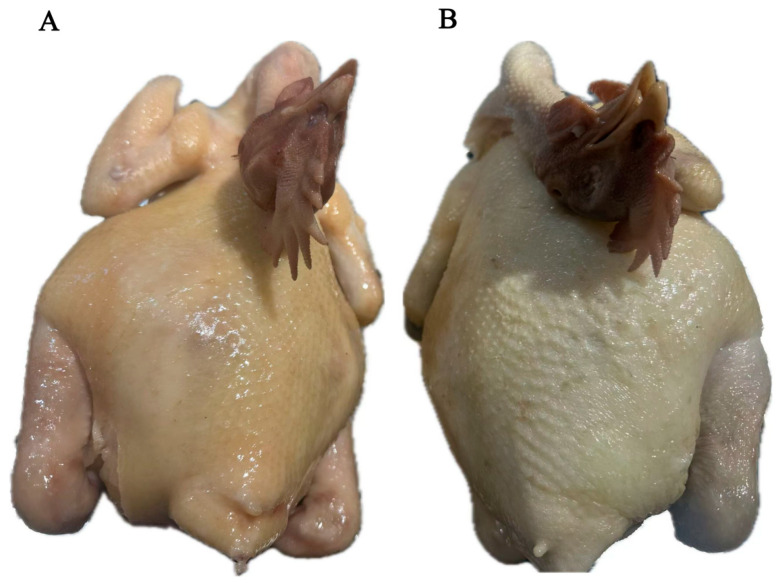
Chicken prepared using different cooking methods. (**A**) PP chicken. (**B**) SW chicken.

**Figure 2 foods-14-03778-f002:**
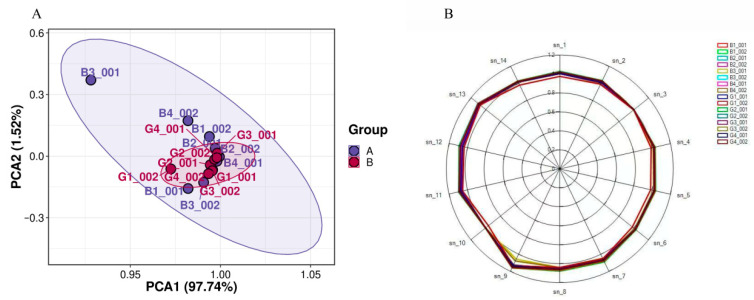
E-nose analysis. (**A**) PCA plot of PP chicken and SW chicken. (**B**) Radar graph of PP chicken and SW chicken.

**Figure 3 foods-14-03778-f003:**
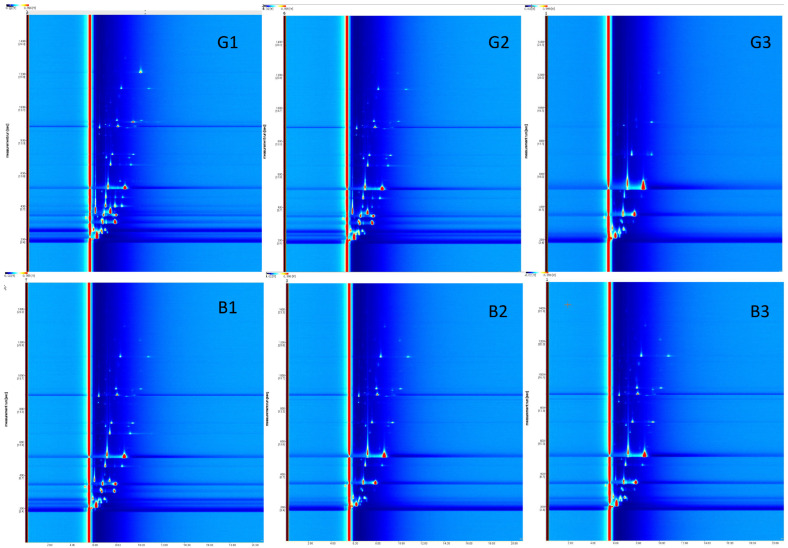
Topographic plots and topographic sub traction plots. Numbers (1–45) indicate the areas (spots, peaks) that are VOCs characterized by the retention time (on the y axis), the relative drift time (on the x axis), and the intensity of the signals. Group G1–3 represents SW chicken; Group B1–3 represents PP chicken.

**Figure 4 foods-14-03778-f004:**
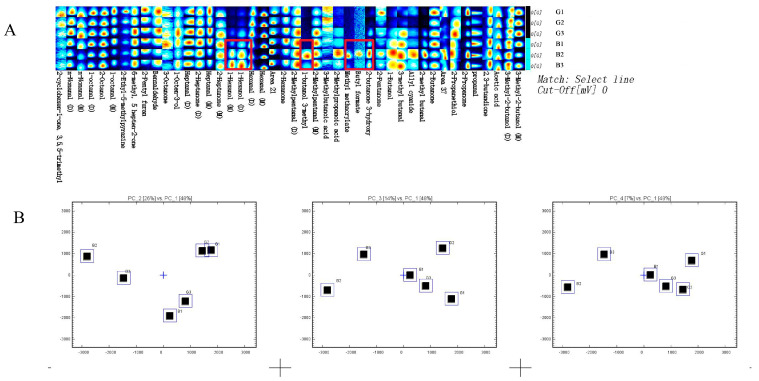
VOCs analysis by HS-GC-IMS. (**A**) VOCs fingerprint comparisons of PP chicken and SW chicken. The redder the area is, the larger is the quantity of VOCs. Each row represents all the signals selected in a sample. Each column represents the signals of the same VOCs. (M) and (D) denote monomer and dimer, respectively. Red boxes indicate VOCs showing significant differences. (**B**) Principal component analysis of VOCs in PP chicken and SW chicken.

**Table 1 foods-14-03778-t001:** Sensory Attributes of Steamed and boiled Wenchang Chicken Evaluated.

Sample	Evaluation Dimension	Specific Indicators	SW Chicken	PP Chicken	*p*-Values
Breast Muscle	Color	Brightness of skin color	4 ± 0.19	3.59 ± 0.17	0.05
		Uniformity of muscle color	3.78 ± 0.22	3.41 ± 0.23	0.11
	Taste	Fresh and savory chicken flavor	3.83 ± 0.25	3.6 ± 0.26	0.33
		Uniformity of seasoning penetration	3.5 ± 0.13	3.38 ± 0.13	0.28
		Persistence of aftertaste	3.33 ± 0.21	3.4 ± 0.26	0.75
		Sweetness perception	3.18 ± 0.17	3.11 ± 0.11	0.56
		Saltiness penetration	3.29 ± 0.13	3.15 ± 0.06	0.15
	Tenderness	Crispness/elasticity of chicken skin	3.33 ± 0.23	3.48 ± 0.26	0.50
		Ease of chewing chicken muscle fibers	3.15 ± 0.06	3.33 ± 0.01	0.01
		Juice retention rate	3.26 ± 0.26	3.15 ± 0.06	0.50

**Table 2 foods-14-03778-t002:** The impact of different methods on skin color.

Indicator		Dorsal Skin		*p*-Values
		PP Chicken	SW Chicken	
Skin color	Brightness value	50.88 ± 5.35	61.05 ± 4.37	4.0 × 10^−6^
	Redness value	1.64 ± 0.8	1.3 ± 0.72	0.23
	Yellowness value	17.21 ± 1.56	12.56 ± 1.76	2.5 × 10^−8^

**Table 3 foods-14-03778-t003:** Comparisons of the detected VOCs in PP chicken and SW chicken by HS-SPME-GC-MS.

Cas	Compound Name	Molecular Formula	Estimated Concentration (μg/kg)			
			PP Chicken	SW Chicken	*p*-Values	Food Change
111-71-7	Heptanal	C_7_H_14_O	44.87 ± 30.45	24.18 ± 8.33	0.36	0.86
139185-79-8	Z,Z-2,5-Pentadecadien-1-ol	C_15_H_28_O	0.04 ± 0.01	0.28 ± 0.12	0.08	−0.85
13019-20-0	3-Heptanone, 2-methyl-	C_8_H_16_O	270.67 ± 0	270.67 ± 0	0.00	0.00
10544-96-4	Octadecane, 6-methyl-	C_19_H_4_0	2.14 ± 1.1	0.6 ± 0.94	0.14	2.54
110-13-4	2,5-Hexanedione	C_6_H_10_O2	24.63 ± 32.42	43.68 ± 30.06	0.50	−0.44
124-13-0	Octanal	C_8_H_16_O	21.01 ± 10.63	21.4 ± 9.33	0.96	−0.02
18409-17-1	2-Octen-1-ol, (E)-	C8H16O	0.91 ± 1.54	1.23 ± 1.81	0.81	−0.26
116465-18-0	2-Trifluoroacetoxytridecane	C_15_H_27_F_3_O_2_	0.12 ± 0.16	0.97 ± 1.06	0.30	−0.88
14905-56-7	Tetradecane, 2,6,10-trimethyl-	C_17_H_36_	0.96 ± 0.39	1.68 ± 0.82	0.27	−0.43
124-19-6	Nonanal	C_9_H_18_O	40.8 ± 17.78	45.31 ± 27.63	0.83	−0.10
112-40-3	Dodecane	C_12_H_26_	3.15 ± 0.3	3.71 ± 2.57	0.74	−0.15
112-31-2	Decanal	C_10_H_20_O	4.64 ± 0.88	4.62 ± 2.89	0.99	0.00
150-86-7	Phytol	C_20_H_40_O	0.47 ± 0.35	0.56 ± 0.62	0.84	−0.16
540-97-6	Cyclohexasiloxane, dodecamethyl-	C_12_H_36_O_6_Si_6_	20.52 ± 9.11	20.11 ± 9.01	0.96	0.02
107-50-6	Cycloheptasiloxane, tetradecamethyl-	C_14_H_42_O_7_Si_7_	3.68 ± 0.56	9.46 ± 12.22	0.41	−0.61
330455-64-6	Thymol, TBDMS derivative	C_16_H_28_OSi	0.1 ± 0.14	0.04 ± 0.04	0.54	1.43
1894-68-4	2-Trifluoroacetoxydodecane	C_14_H_25_F_3_O_2_	0.24 ± 0.24	0.16 ± 0.2	0.69	0.49
66-25-1	Hexanal	C_6_H_12_O	660.81 ± 369.67	911 ± 389.99	0.47	−0.27
3391-86-4	1-Octen-3-ol	C_8_H_16_O	92.25 ± 91.31	49.28 ± 27.85	0.51	0.87
1940-18-7	Cyclohexanol, 1-ethyl-	C_8_H_16_O	3.04 ± 2.61	1.62 ± 1.58	0.48	0.87
2548-87-0	2-Octenal, (E)-	C_8_H_14_O	4.64 ± 4.53	5.28 ± 4.29	0.87	−0.12
541-02-6	Cyclopentasiloxane, decamethyl-	C_10_H_30_O_5_Si_5_	14.87 ± 1.05	29.83 ± 10.44	0.13	−0.50
195194-80-0	2-Piperidinone, N-[4-bromo-n-butyl]-	C_9_H_16_BrNO	0.08 ± 0.07	0.21 ± 0.24	0.38	−0.61
2425-77-6	1-Decanol, 2-hexyl-	C_16_H_34_O	0.95 ± 0.77	0.8 ± 0.74	0.82	0.18
630-08-0	Carbon monoxide	CO	0.14 ± 0.06	0.13 ± 0.04	0.74	0.12
13019-16-4	2-Octenal, 2-butyl-	C_12_H_22_O	6.21 ± 5.77	6.7 ± 7.93	0.93	−0.07
19780-11-1	2-Dodecen-1-yl(-)succinic anhydride	C_16_H_26_O_3_	1.1 ± 0.89	8.35 ± 9.47	0.32	−0.87
1654-86-0	Decanoic acid, decyl ester	C_20_H_40_O_2_	8.48 ± 6.83	4.08 ± 3.11	0.31	1.08
25144-04-1	Cyclopentanol, 2-methyl-, trans-	C_6_H_12_O	0.11 ± 0.11	0.14 ± 0.17	0.74	−0.26
28023-80-5	N-Isopentyl-N-nitroso-pentylamine	C_10_H_22_N_2_O	7.54 ± 7.18	2.64 ± 3.65	0.36	1.86
2122-26-1	Aspidospermidin-17-ol	C_23_H_30_N_2_O_5_	0.34 ± 0.56	0.41 ± 0.19	0.84	−0.16
3555-45-1	Silicic acid,	C_10_H_28_O_4_Si_3_	0.02 ± 0.02	0.03 ± 0.02	0.50	−0.37
18030-67-6	3-Ethoxy-1,	C_11_H_32_O_4_Si_4_	0.02 ± 0.01	0.03 ± 0.04	0.66	−0.31
19095-23-9	Heptasiloxane	C_14_H_44_O_6_Si_7_	0.17 ± 0.17	0.06 ± 0.05	0.37	2.05
150304-08-8	4-Hydroxy-2,2′,4′	C_12_H_6_C_l4_O	0.05 ± 0.05	0.07 ± 0.04	0.57	−0.29
105037-97-6	Azetidine-2-one	C_10_H_19_NO	0.11 ± 0.02	0.65 ± 0.6	0.17	−0.83
29812-79-1	Hydroxylamine,	C_10_H_23_NO	1.31 ± 1.39	2.57 ± 2.15	0.45	−0.49
10537-47-0	Propanedinitrile	C_18_H_22_N_2_O	1.06 ± 1.03	0.65 ± 0.91	0.63	0.63
56797-40-1	7-Hexadecenal, (Z)-	C_16_H_30_O	0.5 ± 0.68	0.24 ± 0.4	0.60	1.12
111-87-5	1-Octanol	C_8_H_18_O	6.45 ± 5.35	4.69 ± 5.05	0.74	0.38
55162-61-3	Tetracontane, 3,5,24-trimethyl-	C_43_H_88_	2.55 ± 4.1	1.13 ± 1.15	0.56	1.26
3891-98-3	Dodecane, 2,6,10-trimethyl-	C_15_H_32_	6.49 ± 6.56	4.06 ± 6.01	0.66	0.60
110-62-3	Pentanal	C_5_H_10_O	58.22 ± 39.61	31.67 ± 10.5	0.37	0.84
55334-42-4	Dodecane, 1,2-dibromo-	C_12_H_24_Br_2_	0.8 ± 0.73	6.1 ± 4.42	0.17	−0.87
74708-73-9	1,4-Methanobenzocyclodecene,	C_15_H_22_	3.47 ± 0.66	12.1 ± 9.27	0.25	−0.71
10552-94-0	1H-Pyrrole, 2,5-dihydro-1-nitroso-	C_4_H_6_N_2_O	0.06 ± 0.02	14.75 ± 29.21	0.39	−1.00
1883-13-2	Dodecanoic acid, 3-hydroxy-	C_12_H_24_O_3_	0.2 ± 0.26	0.13 ± 0.11	0.73	0.49
19095-24-0	Octasiloxane,	C_16_H_50_O_7_Si_8_	0.25 ± 0.19	0.14 ± 0.1	0.42	0.83
87867-97-8	3-Butoxy-1,	C_13_H_36_O_4_Si_4_	0.05 ± 0.05	0.04 ± 0.03	0.86	0.21
15399-43-6	Olean-12-ene-3,15,16,21,22,28-hexol,	C_30_H_50_O_6_	0.04 ± 0.03	0.08 ± 0.05	0.27	−0.53
103577-45-3	Lansoprazole	C_16_H_14_F_3_N_3_O_2_S	0.11 ± 0.09	0.11 ± 0.08	0.97	0.02
5638-09-5	Cyclopentane, (4-octyldodecyl)-	C_25_H_50_	1.32 ± 1.16	5.27 ± 8.67	0.51	−0.75
948-60-7	Pterin-6-carboxylic acid	C_7_H_5_N_5_O_3_	0.05 ± 0.05	0.05 ± 0.05	0.84	0.22
122-16-7	Sulfanitran	C_14_H_13_N_3_O_5_S	37.68 ± 65.15	73.11 ± 112.55	0.67	−0.48
13463-39-3	Nickel tetracarbonyl	C_4_NiO_4_	8.42 ± 14.49	1.5 ± 2.41	0.50	4.63
995-82-4	Hexasiloxane	C_12_H_38_O_5_Si_6_	0.12 ± 0.13	0.02 ± 0.03	0.32	4.17
25144-05-2	Cyclopentanol	C_6_H_12_O	0.03 ± 0.02	0.36 ± 0.58	0.43	−0.91
626-33-5	4-Heptanone	C_8_H_16_O	0.13 ± 0.03	0.29 ± 0.46	0.62	−0.54
624-42-0	3-Heptanone	C_8_H_16_O	0.01 ± 0.01	0.11 ± 0.04	0.03	−0.86
7325-84-0	Silane, trichlorodocosyl-	C_22_H_45_Cl_3_Si	0	0.27 ± 0.22	0.00	#DIV/0!
52132-58-8	Acetic acid, chloro-	C_18_H_35_ClO_2_	1.54 ± 0.07	1.01 ± 0.89	0.41	0.52
1560-96-9	Tridecane	C_14_H_30_	1.83 ± 0.32	10.31 ± 11.02	0.31	−0.82
7225-68-5	Dodecane	C_25_H_48_	0.54 ± 0.48	9.85 ± 14.18	0.37	−0.95
59426-46-9	2,5-Furandione	C_16_H_26_O_3_	0.5 ± 0.26	1.47 ± 0.28	0.04	−0.66
3892-00-0	Pentadecane	C_18_H_38_	1.62 ± 0.39	12.74 ± 14.65	0.32	−0.87
25360-09-2	tert-Hexadecanethiol	C_16_H_34_S	2.49 ± 2.57	6.52 ± 6.2	0.39	−0.62
5057-99-8	1,2-Cyclopentanediol	C_5_H_10_O_2_	0.09 ± 0.11	7.45 ± 7.14	0.22	−0.99
112-86-7	Erucic acid	C_22_H_42_O_2_	1.13 ± 1.35	0.89 ± 0.89	0.85	0.28
10584-64-2	D-Homo-24-nor-17-oxachola-20,22-diene-3,7,16-trione, 14,15:21,23-diepoxy-4,4,8-trimethyl-	C_26_H_32_O_6_	0.04 ± 0.05	0.07 ± 0.04	0.54	−0.42
56769-23-4	trans-3,4-Epoxynonane	C_9_H_18_O	0.06 ± 0.07	0.05 ± 0.04	0.85	0.24
1428-33-7	3-(1,1,2,2-Tetrafluoroethoxy)prop-1-ene	C_5_H_6_F_4_O	0	0.24 ± 0.33	0.00	N/A
29783-26-4	4-Cyclopentene-1,3-diol, cis-	C_5_H_8_O_2_	0	8.13 ± 5.3	0.00	N/A
1068-57-1	Acetic acid, hydrazide	C_2_H_6_N_2_O	0	29.63 ± 51.28	0.00	N/A
60609-53-2	8-Hexadecenal, 14-methyl-, (Z)-	C_17_H_32_O	0	1.46 ± 0.89	0.00	N/A
55255-85-1	Cyclopentane, 1,1′-[3-(2-cyclopentylethyl)-1,5-pentanediyl]bis-	C_22_H4_0_	0	1.05 ± 1.47	0.00	N/A
6982-39-4	trans-2-Aminocyclohexanol	C_6_H_13_NO	0	3.55 ± 5.12	0.00	N/A
38520-31-9	Oxiraneundecanoic acid, 3-pentyl-, methyl ester, trans-	C_19_H_36_O_3_	0.24 ± 0.23	0	0.00	N/A
29804-22-6	Disparlure	C_19_H_38_O	1.28 ± 0.98	0	0.00	N/A
143-08-8	1-Nonanol	C_9_H_20_O	0	0.02 ± 0.01	0.00	N/A
7045-79-6	Oxetane, 2-methyl-4-propyl-	C_7_H_14_O	0	0.06 ± 0.04	0.00	N/A

4-Heptanone, 2-methyl and 3-Heptanone, 6-methyl are ketone compounds typically associated with fruity or creamy aromas, likely formed through the degradation of proteins and fats under high-temperature conditions. In SW chicken, where the cooking temperature is higher, fat oxidation may be more pronounced, leading to higher concentrations of these ketones, whereas in PP chicken, which is prepared at a lower temperature, their levels may be lower. 1,2-Cyclopentanediol, trans- and 4-Cyclopentene-1,3-diol, cis- are diol compounds possibly derived from fatty acid degradation or complex thermal reactions. They have a slight sweet aroma and may influence the sensory characteristics of chicken meat. Their presence indicates differences in the pathways of fat degradation between the two cooking methods. 8-Hexadecenal, 14-methyl-, (Z)-, an aldehyde compound with a strong fruity and fatty aroma, is a key product of fatty acid degradation. Accelerated fat oxidation under high temperatures likely contributes to its higher concentration in SW chicken compared to PP chicken. Disparlure, although a component of pheromones, may be detected in food due to specific microbial activity or unique fat degradation reactions. Its sweet and floral notes may play a unique role in the aroma profile of chicken. 1-Nonanol, a soft floral alcohol, is produced through fat oxidation or alcohol metabolism. It not only enhances the aroma complexity of chicken but also reflects the impact of different cooking methods on flavor formation. The distinct temperatures used in preparing boiled and SW chicken significantly influence the generation of VOCs. High-temperature environments promote the formation of specific aroma compounds through fat oxidation, Maillard reactions, and amino acid degradation [15], while lower temperatures may better preserve heat-sensitive compounds [16]. These unique VOCs provide potential markers to differentiate the muscle characteristics of the two chicken preparation methods.

## Data Availability

The original contributions presented in this study are included in the article. Further inquiries can be directed to the corresponding author.

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
