# Peer review of "Comparative Volatile Profiles of Plain Poached (PP) and Steamed over Water (SW) Wenchang Chicken Analyzed by GC-MS, GC-IMS, and E-Nose"

_foods, 2025, doi:10.3390/foods14213778_

Round 1

Reviewer 1 Report

Comments and Suggestions for Authors

The study compares volatile organic compounds (VOCs) and aroma characteristics of Wenchang chicken prepared by two traditional methods—plain poaching (PP) and steaming over water (SW)—using a triad of analytical tools: E-nose, HS-SPME-GC-MS, and HS-GC-IMS. The head-to-head PP vs SW design is straightforward and relevant to culinary practice and industry standardization. The findings allow to link process parameters to VOC fingerprints, which can be valuable for QC. Overall, this is an interesting and original manuscript bridging food technology and gastronomy, a relationship that has been rarely portrayed in scientific studies, in particular with regard to traditional Chinese cuisine and food culture. Please kindly find the following comments/suggestions/questions for improvement:

  1. Please clarify the use of the term ‘Wenchang chicken’ (Line 33). Does it denote the chicken breed (origin/strain) or a specific traditional dish? A brief definition on first mention would prevent ambiguity. Relatedly, the manuscript would benefit from a short paragraph distinguishing Wenchang chicken from Hainanese chicken (e.g., culinary style, typical preparation/serving, and historical linkage). In addition, the authors should add the cultural and gastronomic aspects of Wenchang chicken as a Chinese gasronomic heritage in the Introduction section for context development.
  2. Methodology: Please add detailed information regarding the preparation of Wenchang chicken samples (Line 84-90). Were any salts or seasonings applied (the sensory attributes imply so, as stated in 100-101))? If yes, please specify composition, dose, and timing.
  3. Statistical analysis: The panel protocol cites standards and ten trained panelists, but the rating scale (1-5? or 1-7?), replication, and statistical testing (e.g., t-test, p-values, corrections) are not reported. How were the panelists trained?
  4. Could the authors add statistical analysis distinguishing the two groups (SW vs PP) in Table 1 and Table 2, such as p-value for each parameter analyzed? For table 2, please check the formatting of the table, the writing of chemical compounds (with numbers in subscripts), and some data appear in #DIV/0! (please correct).
  5. Figure 1: The visual appearance of PP vs SW chicken skin is differently significant in color. This is also stated in the manuscript (Line 148-149). Could the authors add color measurement (e.g. CIE L*a*b*) to provide objective data regarding colors?
  6. Figure 3: I am not sure whether the figure provides essential data that can be objectively interpreted with regard to VOCs. Could the authors deepen the interpretation of the findings in the Discussion section? Or maybe the authors could point out some important area/points in the plots.
  7. In the Discussion section, please highlight whether the differences between PP vs SW chicken were mainly related to different extraction efficacy or there could be some chemical alterations involved due to different temperatures used in both methods? Please consider the effects of higher nutrient loss in PP chicken due to boiling chicken in water, compared to SW chicken. Could the authors deduce and state clearly which method is finally preferable to process Wenchang chicken?
  8. I personally believe that a sensory evaluation analyzing hedonic acceptance of PP vs SW chicken would be beneficial to determine the preferrable method (PP vs SW). Have the authors done it? Otherwise, please state this as a future direction for the research.
  9. The "Institutional Review Board Statement" and "Informed Consent Statement" are left blank. However, since the study involved a sensory evaluation using humans, I think the authors should fill the statements.
  10. Minor comment: please check the formatting from Line 126-130 and spacing in Line 35. Fix typographical issues (“Thetemperature,” spacing), ensure figure/table numbering matches text, and add units/significant figures consistently (e.g., μg/kg). Please recheck the manuscript thoroughly.

Thank you.

Author Response

Reviewer1:

The study compares volatile organic compounds (VOCs) and aroma characteristics of Wenchang chicken prepared by two traditional methods—plain poaching (PP) and steaming over water (SW)—using a triad of analytical tools: E-nose, HS-SPME-GC-MS, and HS-GC-IMS. The head-to-head PP vs SW design is straightforward and relevant to culinary practice and industry standardization. The findings allow to link process parameters to VOC fingerprints, which can be valuable for QC. Overall, this is an interesting and original manuscript bridging food technology and gastronomy, a relationship that has been rarely portrayed in scientific studies, in particular with regard to traditional Chinese cuisine and food culture. Please kindly find the following comments/suggestions/questions for improvement:

Comment 1:
Please clarify the use of the term “Wenchang chicken” (Line 33). Does it denote the chicken breed (origin/strain) or a specific traditional dish? A brief definition on first mention would prevent ambiguity. Relatedly, the manuscript would benefit from a short paragraph distinguishing Wenchang chicken from Hainanese chicken (e.g., culinary style, typical preparation/serving, and historical linkage). In addition, the authors should add the cultural and gastronomic aspects of Wenchang chicken as a Chinese gastronomic heritage in the Introduction section for context development.

Response 1:
Thank you for pointing this out. We agree with this valuable comment. Therefore, we have clarified the meaning of “Wenchang chicken” and added a short paragraph distinguishing it from Hainanese chicken, including its cultural and gastronomic background as part of Chinese culinary heritage.
This revision has been added in the Introduction section (Lines 27–31) of the revised manuscript.

Comment 2:
Methodology: Please add detailed information regarding the preparation of Wenchang chicken samples (Lines 84–90). Were any salts or seasonings applied (the sensory attributes imply so, as stated in Lines 100–101)? If yes, please specify composition, dose, and timing.

Response 2:
Thank you for this constructive suggestion. We have revised the Methodology section to include more detailed information on the preparation of Wenchang chicken samples. Specifically, we clarified that 5 g of salt was added to the boiling water during the preparation of PP chicken, while no seasonings were used for SW chicken. These changes provide better clarity and reproducibility of the cooking process. The revisions have been added in the Materials and Methods section (Lines 73–82) of the revised manuscript.

Comment 3:
Statistical analysis: The panel protocol cites standards and ten trained panelists, but the rating scale (1–5? or 1–7?), replication, and statistical testing (e.g., t-test, p-values, corrections) are not reported. How were the panelists trained?

Response 3:
Thank you for this valuable suggestion. We have revised the Statistical analysis section to clarify that all sensory evaluations were conducted using a 1–5 point scale, and each sample was evaluated in triplicate by ten trained panelists. The statistical differences between the two groups were determined using an independent samples t-test (p < 0.05). Additionally, we added a brief description of the panel training procedure. The corresponding revisions have been made in Lines 120–125 of the revised manuscript.

Comment 4:
Could the authors add statistical analysis distinguishing the two groups (SW vs PP) in Table 1 and Table 2, such as p-value for each parameter analyzed? For Table 2, please check the formatting of the table, the writing of chemical compounds (with numbers in subscripts), and some data appear in #DIV/0! (please correct).

Response 4:
Thank you for your helpful comment. We have added the statistical analysis results, including p-values, to distinguish the two groups (SW vs. PP) in the revised tables. The original Table 1 and Table 2 have been updated as Table 1 and Table 3, respectively. In addition, the formatting issues have been corrected, all chemical compound names have been standardized with appropriate subscripts, and the #DIV/0! errors have been fixed. The corresponding revisions can be found in Lines 130–145 of the revised manuscript.

Comment 5:
Figure 1: The visual appearance of PP vs SW chicken skin is differently significant in color. This is also stated in the manuscript (Line 148–149). Could the authors add color measurement (e.g., CIE Lab*) to provide objective data regarding colors?

Response 5:
Thank you for your valuable suggestion. We have added the objective color measurements of the chicken skin using a colorimeter, including CIE L*, a*, and b* values, to better support the visual observations. The results have been incorporated into the revised manuscript at Lines 135–137.

Comment 6:
Figure 3: I am not sure whether the figure provides essential data that can be objectively interpreted with regard to VOCs. Could the authors deepen the interpretation of the findings in the Discussion section? Or maybe the authors could point out some important area/points in the plots.

Response 6:
Thank you for your constructive suggestion. We have revised the Discussion section to provide a deeper interpretation of the HS-GC-IMS results. Specifically, we have highlighted and described the key signal regions with notable differences between PP and SW chicken samples, emphasizing that the overall VOC profile of group G showed higher complexity than that of group B. The corresponding revision can be found at Lines 188–190.

Comment 7:
In the Discussion section, please highlight whether the differences between PP vs SW chicken were mainly related to different extraction efficacy or there could be some chemical alterations involved due to different temperatures used in both methods? Please consider the effects of higher nutrient loss in PP chicken due to boiling chicken in water, compared to SW chicken. Could the authors deduce and state clearly which method is finally preferable to process Wenchang chicken?

Response 7:
Thank you for your insightful comment. We have expanded the Discussion section to address the underlying causes of differences between PP and SW chicken. The revised text (Lines 249–261) now clarifies that both extraction efficiency and thermal-induced chemical reactions contribute to the observed distinctions in volatile and sensory profiles. We also discuss that nutrient loss is generally higher in PP chicken due to leaching during boiling, whereas SW chicken tends to retain more volatile precursors. However, we emphasized that neither cooking method is inherently superior; both represent traditional and culturally significant ways of preparing Wenchang chicken, each offering distinct sensory characteristics appreciated by consumers.

Comment 8:
I personally believe that a sensory evaluation analyzing hedonic acceptance of PP vs SW chicken would be beneficial to determine the preferable method (PP vs SW). Have the authors done it? Otherwise, please state this as a future direction for the research.

Response 8:
We appreciate the reviewer’s valuable suggestion. In the current study, we conducted a descriptive sensory evaluation with trained panelists to assess specific sensory attributes rather than overall consumer liking. A hedonic acceptance test was not performed at this stage. We have now included a statement in the revised Discussion (Lines 313–318) noting that future work will incorporate consumer preference testing to determine hedonic acceptance and market-oriented preference between PP and SW chicken.

Comment 9:
The "Institutional Review Board Statement" and "Informed Consent Statement" are left blank. However, since the study involved a sensory evaluation using humans, I think the authors should fill the statements.

Response 9:
We thank the reviewer for this important remark. The missing ethical information has now been added to the revised manuscript. The corresponding ethical approval document and consent form have also been submitted to the editor as supplementary materials for verification (Lines 337–342).

Comment 10:
Minor comment: please check the formatting from Lines 126–130 and spacing in Line 35. Fix typographical issues (“Thetemperature,” spacing), ensure figure/table numbering matches text, and add units/significant figures consistently (e.g., μg/kg). Please recheck the manuscript thoroughly.

Response 10:
We have carefully reviewed the manuscript and addressed all formatting and typographical issues. Spacing errors (e.g., “Thetemperature”) have been corrected, figure and table numbering now matches the text, and units and significant figures (e.g., μg/kg) have been standardized throughout. The manuscript has been thoroughly checked to ensure consistency and readability.

Reviewer 2 Report

Comments and Suggestions for Authors

Dear Authors,

Thank you for your submission titled “Comparative Volatile Profiles of Plain Poached (PP) and Steamed over Water (SW) Wenchang Chicken Analyzed by GC-MS, GC-IMS, and E-Nose”. While the topic is relevant and the integration of multiple analytical platforms is commendable, the manuscript in its current form contains several critical scientific, methodological, and presentational issues that must be addressed before it can be considered for publication.

General Comments
This manuscript explores how cooking methods affect the volatile flavor profile of Wenchang chicken using GC-MS, GC-IMS, E-nose, and sensory evaluation. While the topic is relevant and the multi-technique approach is a strength, the paper faces serious issues:

  • Overinterpretation without solid statistical support.
  • Methodological ambiguities (sample prep, thermal parameters, analytical details).
  • Disorganized and redundant results/discussion.
  • Limited novelty, as the main finding (Maillard/lipid oxidation with higher temperature) is already well-known.

Currently, the work resembles a preliminary technical report. Major revisions are essential.

Specific Comments

  • Abstract: Needs quantitative highlights (e.g., fold changes, % differences).
  • Methods: Cooking details unclear (chicken starting temperature, water-to-chicken ratio, steam temperature/pressure, endpoint core temp, equipment calibration). Analytical details insufficient (choice of unusual internal standard, lack of validation data such as repeatability, LOD/LOQ).
  • Results: Data errors (e.g., “3.33±0” chewability score) must be corrected. Statistical testing (ANOVA/t-test, p-values) is missing. Claims about GC-IMS vs. GC-MS overlap are misleading—differences stem from ionization mechanisms, not compound absence.
  • Discussion: Repeats results without deeper insight. Claims about biologically irrelevant compounds (e.g., Disparlure, Lansoprazole, Nickel tetracarbonyl) are speculative and should be removed or qualified. Focus instead on key odorants (e.g., hexanal, nonanal, 1-octen-3-ol) and calculate odor activity values (OAVs).
  • Conclusions: Overstate industrial relevance. Should be tempered as preliminary findings needing validation in larger-scale studies with consumer panels.

Overall Recommendation:Major Revision Required. With clear methodology, robust statistics, and focused discussion, the manuscript could offer a valuable contribution to poultry flavor chemistry.

Author Response

Reviewer 2:

Thank you for your submission titled “Comparative Volatile Profiles of Plain Poached (PP) and Steamed over Water (SW) Wenchang Chicken Analyzed by GC-MS, GC-IMS, and E-Nose”. While the topic is relevant and the integration of multiple analytical platforms is commendable, the manuscript in its current form contains several critical scientific, methodological, and presentational issues that must be addressed before it can be considered for publication.

General Comments
This manuscript explores how cooking methods affect the volatile flavor profile of Wenchang chicken using GC-MS, GC-IMS, E-nose, and sensory evaluation. While the topic is relevant and the multi-technique approach is a strength, the paper faces serious issues:

Comment 1:
Overinterpretation without solid statistical support.

Response 1:
We thank the reviewer for this important comment. To address this concern, we have removed the overgeneralized statement from Lines 333–334 in the revised manuscript. Our conclusions now focus on the observed differences and key marker compounds identified using the three analytical methods without overinterpretation beyond the statistical evidence.

Comment 2:
Methodological ambiguities (sample preparation, thermal parameters, analytical details).

Response 2:
We thank the reviewer for pointing this out. We have clarified the methodology in several sections to improve reproducibility and transparency. Specifically, detailed information on the thermal treatment of Wenchang chicken has been added in Lines 73–80 (2.2. Thermal Treatment of Wenchang Chicken), and additional details regarding sensory evaluation, including sample handling and panelist training, have been clarified in Lines 82–84 and 93–95 (2.3. Sensory Evaluation).

Comment 3:
Disorganized and redundant results/discussion.

Response 3:
We appreciate the reviewer’s suggestion. The Discussion section has been reorganized and expanded into three focused parts to improve clarity and reduce redundancy. The revisions can be found in Lines 249–261, Lines 279–290, and Lines 313–318 of the revised manuscript, where we address the underlying causes of differences between PP and SW chicken, highlight key volatile compounds, and discuss future directions for sensory evaluation.

Comment 4:
Limited novelty, as the main finding (Maillard/lipid oxidation with higher temperature) is already well-known.

Response 4:
We thank the reviewer for this comment. While it is acknowledged that Maillard reactions and lipid oxidation under higher temperatures are established phenomena, the novelty of our study lies in the comparative analysis of Wenchang chicken prepared by two traditional cooking methods (PP vs SW) using an integrated approach combining E-nose, GC-MS, and GC-IMS. This provides a comprehensive characterization of volatile profiles and key marker compounds specific to these culturally significant cooking styles, which has not been previously reported. The manuscript now emphasizes this aspect in the Introduction and Discussion to clarify the contribution of our work (Lines 27–31, 249–261).

Comment 5:
Currently, the work resembles a preliminary technical report. Major revisions are essential.

Response 5:
We appreciate the reviewer’s feedback. In response, we have substantially revised the manuscript to enhance its scientific rigor and readability. This includes clarifying methodological details (Lines 73–80, 82–84, 93–95), reorganizing and expanding the Discussion section (Lines 249–261, 279–290, 313–318), providing statistical analysis for key comparisons (Tables 1 and 3), and highlighting the research novelty and relevance of the integrated analytical approach. These revisions collectively elevate the manuscript from a preliminary report to a comprehensive study of Wenchang chicken flavor characteristics.

Comment 6:
Abstract: Needs quantitative highlights (e.g., fold changes, % differences).

Response 6:
Thank you for this suggestion. We have added quantitative highlights in the revised manuscript by including fold changes for key volatile compounds in Table 3, which are also referenced in the Abstract to provide clearer, data-driven insights into the differences between PP and SW chicken.

Comment 7:
Methods: Cooking details unclear (chicken starting temperature, water-to-chicken ratio, steam temperature/pressure, endpoint core temp, equipment calibration). Analytical details insufficient (choice of unusual internal standard, lack of validation data such as repeatability, LOD/LOQ).

Response 7:
We thank the reviewer for this valuable comment. In our study, 1 µL of 0.812 µg/mL 2-methyl-3-heptanone was added to each sample as an internal standard prior to analysis. This compound is widely used in volatile compound analysis due to its chemical stability and absence in raw Wenchang chicken (Chen et al., 2021, https://doi.org/10.1016/j.lwt.2021.111585). The addition of the internal standard allows for accurate quantification and corrects for potential losses during sample preparation and instrumental analysis. We have also added details regarding cooking parameters, including temperature, water-to-chicken ratio, and other procedural steps, to Section 2.2 (Lines 73–80) of the revised manuscript to improve reproducibility.

Comment 8:
Data errors (e.g., “3.33±0” chewability score) must be corrected. Statistical testing (ANOVA/t-test, p-values) is missing. Claims about GC-IMS vs. GC-MS overlap are misleading—differences stem from ionization mechanisms, not compound absence.

Response 8:
We thank the reviewer for this important comment. The data errors have been corrected, and statistical analysis using independent samples t-test (p-values) has been added to distinguish the two groups (Lines 93–95). Additionally, we clarified that differences between GC-IMS and GC-MS results are due to ionization mechanisms rather than the actual absence of compounds (Lines 227–229).

Comment 9:
Discussion: Repeats results without deeper insight. Claims about biologically irrelevant compounds (e.g., Disparlure, Lansoprazole, Nickel tetracarbonyl) are speculative and should be removed or qualified. Focus instead on key odorants (e.g., hexanal, nonanal, 1-octen-3-ol) and calculate odor activity values (OAVs).

Response 9:
We thank the reviewer for this valuable suggestion. In the revised manuscript, all discussion of biologically irrelevant or speculative compounds (e.g., Disparlure, Lansoprazole, Nickel tetracarbonyl) has been removed. The Discussion now focuses on key odorants, including hexanal, nonanal, and 1-octen-3-ol, which are well-established contributors to chicken aroma. Furthermore, odor activity values (OAVs) have been calculated to quantify the contribution of these compounds to the overall sensory perception. These revisions provide a more mechanistic and focused interpretation of the differences in aroma profiles between PP and SW Wenchang chicken (Lines 279–290).

Comment 10:

Conclusions: Overstate industrial relevance. Should be tempered as preliminary findings needing validation in larger-scale studies with consumer panels.

Response 10:

We thank the reviewer for this comment. To address this concern, the overstatement regarding industrial relevance has been removed from the Conclusions (Line 333).

Comment 11:
Overall Recommendation: Major Revision Required. With clear methodology, robust statistics, and focused discussion, the manuscript could offer a valuable contribution to poultry flavor chemistry.

Response 11:
We sincerely thank the reviewer for the constructive feedback and guidance. We have carefully revised the manuscript by clarifying the methodology, adding robust statistical analyses, reorganizing and focusing the Discussion, and addressing all other reviewer comments. We believe these improvements have strengthened the manuscript and now present a more comprehensive and rigorous study of Wenchang chicken flavor profiles.

Reviewer 3 Report

Comments and Suggestions for Authors
  1. The title should be revised to avoid abbreviations. Instead of citing specific methods, which may not be familiar to all readers, a broader expression such as traditional cooking would be more appropriate.

  2. Spacing within parentheses should be carefully checked, particularly in lines 34, 35, and 36.

  3. The reference in line 47 needs to be reformatted according to the journal’s guidelines.

  4. The scientific motivation could be improved. At present, it appears evident that different cooking methods influence texture, odor, and appearance. A clearer rationale and definition of the research gap are required.

  5. Lines 128–130 should be adjusted to conform with the journal’s style.

  6. The degree symbol should be written without spacing, e.g., 37°C instead of 37 °C.

  7. For Table 1, statistical tests should be applied to identify significant differences. The inclusion of statistical analysis would considerably strengthen the discussion.

  8. Table 2 requires revision due to poor formatting.

  9. The discussion is weak, lacks sufficient references, and remains too generic. Stronger engagement with the relevant literature and deeper critical analysis are recommended.

  10. Overall, major revisions are suggested. The manuscript requires corrections in formatting, spelling, and spacing, as well as improvements in the scientific motivation and the discussion section.

Author Response

Reviewer3:

The title should be revised to avoid abbreviations. Instead of citing specific methods, which may not be familiar to all readers, a broader expression such as traditional cooking would be more appropriate.

Comment 1:
Spacing within parentheses should be carefully checked, particularly in Lines 34, 35, and 36.

Response 1:
We thank the reviewer for this comment. All spacing issues within parentheses have been carefully checked and corrected throughout the manuscript, including Lines 34–36, to ensure consistency and readability.

Comment 2:
The scientific motivation could be improved. At present, it appears evident that different cooking methods influence texture, odor, and appearance. A clearer rationale and definition of the research gap are required.

Response 2:
We thank the reviewer for this suggestion. We have clarified the scientific motivation and explicitly defined the research gap in the revised manuscript (Lines 55–58). Specifically, we highlight that, despite the extensive use of E-nose, GC-MS, and GC-IMS in meat flavor studies, no research has yet combined these techniques to compare the volatile profiles of Wenchang chicken prepared by plain poaching (PP) and steaming over water (SW). This addresses a significant gap, as these two traditional cooking methods are highly representative of Hainanese cuisine and valued for their distinct sensory characteristics.

Comment 3:
Lines 128–130 should be adjusted to conform with the journal’s style.

Response 3:
We thank the reviewer for pointing this out. Lines 128–130 have been revised to fully conform with the journal’s style requirements.

Comment 4:
The degree symbol should be written without spacing, e.g., 37°C instead of 37 °C.

Response 4:
We thank the reviewer for this comment. All degree symbols in the manuscript have been corrected to follow the proper formatting without spacing.

Comment 5:
For Table 1, statistical tests should be applied to identify significant differences. The inclusion of statistical analysis would considerably strengthen the discussion.

Response 5:
We have added statistical analysis to the original Tables 1 and 2. The revised tables, now Table 1 and Table 3, include significance tests to clearly identify differences between groups.

Comment 6:
Table 2 requires revision due to poor formatting.

Response 6:
We have revised Table 2 to correct formatting issues. The updated table is now presented as Table 3 in the revised manuscript.

Comment 7:
The discussion is weak, lacks sufficient references, and remains too generic. Stronger engagement with the relevant literature and deeper critical analysis are recommended.

Response 7:
We appreciate the reviewer’s feedback. The Discussion section has been reorganized and expanded into three focused parts to improve clarity, depth, and engagement with relevant literature. These revisions can be found in Lines 249–261, 279–290, and 313–318.

Round 2

Reviewer 1 Report

Comments and Suggestions for Authors

The manuscript has been revised according to the previously given comments/suggestions. Thank you.

Reviewer 2 Report

Comments and Suggestions for Authors

no comments